# PANOGEN: Text-Conditioned Panoramic Environment Generation for Vision-and-Language Navigation

**Jialu Li**    **Mohit Bansal**
UNC Chapel Hill
{jialuli, mbansal}@cs.unc.edu
https://pano-gen.github.io

## Abstract

Vision-and-Language Navigation requires the agent to follow language instructions to navigate through 3D environments. One main challenge in Vision-and-Language Navigation is the limited availability of photorealistic training environments, which makes it hard to generalize to new and unseen environments. To address this problem, we propose PANOGEN, a generation method that can potentially create an infinite number of diverse panoramic environments conditioned on text. Specifically, we collect room descriptions by captioning the room images in existing Matterport3D environments, and leverage a state-of-the-art text-to-image diffusion model to generate the new panoramic environments. We use recursive outpainting over the generated images to create consistent 360-degree panorama views. Our new panoramic environments share similar semantic information with the original environments by conditioning on text descriptions, which ensures the co-occurrence of objects in the panorama follows human intuition, and creates enough diversity in room appearance and layout with image outpainting. Lastly, we explore two ways of utilizing PANOGEN in VLN pre-training and fine-tuning. We generate instructions for paths in our PANOGEN environments with a speaker built on a pre-trained vision-and-language model for VLN pre-training, and augment the visual observation with our panoramic environments during agents' fine-tuning to avoid overfitting to seen environments. Empirically, learning with our PANOGEN environments achieves the new state-of-the-art on the Room-to-Room, Room-for-Room, and CVDN datasets. Besides, we find that pre-training with our PANOGEN speaker data is especially effective for CVDN, which has under-specified instructions and needs commonsense knowledge to reach the target. Lastly, we show that the agent can benefit from training with more generated panoramic environments, suggesting promising results for scaling up the PANOGEN environments to enhance agents' generalization to unseen environments.

## 1 Introduction

Vision-and-Language Navigation (VLN) requires an agent to make sequential decisions based on both language instructions and visual environments. In indoor instruction-guided navigation, many datasets have been proposed. These datasets aim to enhance agents' ability to understand detailed navigation instructions [2], dialogue style instructions [52], instructions in different languages [26], and high-level object-finding instructions [43]. Though many efforts have been proposed to help the agent learn diverse instruction inputs, all these datasets are built on the same 3D room environments from Matterport3D, which only contains 60 different room environments for agents' training. This is because diverse photorealistic 3D room environments with a large number of sampled human-height viewpoints are very hard to collect. This limited availability of training environments poses challenges for the agent to learn the navigation policy well, and generalize to unseen new room environments.

37th Conference on Neural Information Processing Systems (NeurIPS 2023).

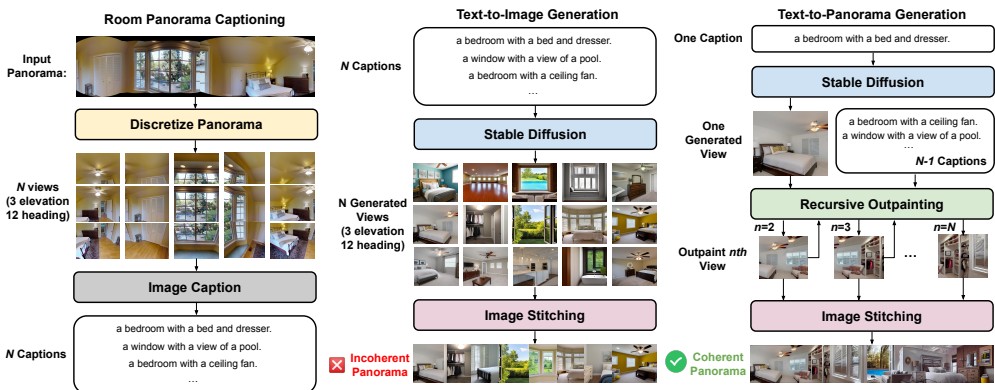

Figure 1: Overview of our PANOGEN. We first generate captions for all the room panoramas in the Matterport3D dataset. Each panorama is discretized into 36 views, we show 15 views here for a better view of each discretized image. Then, we generate panoramic environments with recursive outpainting over a single image generated from the text caption.

Several works in VLN have been proposed to address this challenge. [51] first proposes to perform dropout on environments during training to avoid the agent overfitting to seen environments. [34] and [31] further propose to edit the existing environments by mixing up environments and changing the style and appearance of the environments. However, these approaches are still limited by the 3D environments in Matterport3D, and do not create new environments with different objects and layouts, which is important for agents' generalization to unseen environments. [11] takes one step further and introduces more unannotated 3D room environments from Habitat Matterport3D dataset (HM3D) [46], with machine-generated instructions to augment the training environments and enhance agents' generalization performance. While 900 environments are introduced, their method still relies on existing manually-captured 3D environments in HM3D, which cannot be further scaled up due to the very expensive environment collection process. Hence, in this paper, we aim to explore the possibility of generating unlimited panoramic environments without any human annotation, and investigate how to effectively utilize the generated panoramic environments for improving navigation agents' ability to generalize to unseen environments.

To this end, we propose PANOGEN, a generation method that can potentially create infinite diverse panoramic environments conditioned on text. As shown in Figure 1, we first collect descriptions of room environments by using a state-of-the-art vision-language model BLIP-2 [32] to annotate the view images in the Matterport3D dataset. Then, we use text-to-image diffusion models to generate diverse room images based on the text captions. As the agent navigates in egocentric panorama observation, learning from disjoint single-view images (middle column in Figure 1) will confuse the agent, and it cannot learn the spatial relationship between objects due to inconsistent views. Hence, to keep the observations coherent in the same panorama, we additionally propose a recursive image outpainting approach, which generates missing observations beyond the original image boundaries (right column in Figure 1). Specifically, we choose one generated image in the panorama as the starting point, and gradually rotate the camera angle right, left, up, and down, and then outpaint the unseen observation based on text descriptions. Lastly, we explore and compare two training methods to effectively utilize the generated panoramic environments. In the first method, we train a speaker to automatically generate instructions for the generated panoramic environments, based on a pre-trained vision-and-language model mPLUG [27], which has the state-of-the-art performance for zero-shot video captioning. We pre-train the VLN agent with both the original training data and the generated instruction-trajectory pairs from our panoramic environments. In the second method, instead of learning from speaker data, we directly fine-tune the VLN agents on both the original environments and our panoramic environments by randomly replacing some of the original observations with our panoramic environments to avoid overfitting to training environments.

We conduct experiments on Room-to-Room (R2R) [2], Room-for-Room (R4R) [21], and CVDN [52] datasets. We measure agents' ability in following fine-grained instructions of various lengths (R2R, and R4R), and under-specified dialogue instructions (CVDN). Empirical results demonstrate that training with our PANOGEN environments could improve the SotA agents by 2.7% in success rate and 1.9% in SPL on the R2R test leaderboard, and improve the goal progress by 1.59 meters on the CVDN

test leaderboard, achieving a relative gain of 28.5% compared with previous SotA agents. The large improvement on the CVDN dataset suggests that our PANOGEN introduces diverse commonsense knowledge of the room environments, and helps navigation when given under-specified dialogue instructions. Moreover, we analyze the impact of the number of our PANOGEN environments used for VLN training, and demonstrate the potential of scaling up our PANOGEN environments for further enhancing agents' generalization ability. Lastly, we use both qualitative and quantitative analysis to demonstrate good alignment between our generated instructions and PANOGEN environments.

## 2  Related Work

**Vision-and-Language Navigation** Vision-and-Language Navigation is the task that requires an agent to navigate through the visual environment based on language instructions. Many datasets [2, 21, 52, 8, 39, 49, 41, 26, 43] have been proposed for this task. In this paper, we focus on indoor navigation, specifically Room-to-Room dataset (R2R), Room-four-Room dataset (R4R), and Cooperative Vision-and-Dialog Navigation dataset (CVDN). To solve this challenging task, early works build their model based on a sequence-to-sequence LSTM model [51, 35, 29, 30, 55, 33]. [36] first utilizes pre-trained vision-and-language transformers for learning an agent to pick the correct path. [20] enhances the transformer-based agent with a recurrent state to model navigation history, and [10] proposes a hierarchical architecture to encode both spatial and temporal information in navigation history. Besides, some works explore utilizing graph information to build better global visual representation [12, 53, 63, 15, 9], some works propose better proxy tasks for learning better temporal knowledge [44] and decision making ability [10, 28] during pre-training, and other works explore enriching the navigation agent with the ability to generate text [56] and image [28]. In this paper, we build our approach over the state-of-the-art agent DUET [12].

**Environment Scarcity in Vision-and-Language Navigation** One main challenge in Vision-and-Language Navigation is to learn from limited available training environments and generalize to unseen new environments. Though many datasets have been proposed for indoor instruction-guided navigation with diverse instruction inputs [2, 26, 21, 52, 43], their navigation environments are all from Matterport3D [7], which only contains 61 environments for training, and 29 environments for unseen validation and testing. Previous works aim to address this challenge from multiple aspects. One line of work tries to mitigate the environment bias during training by dropping out environment features [51] and learning environment-agnostic visual representation [30, 57]. Another line of research aims to generate or introduce more environments for VLN training. Specifically, [31] and [34] edit the existing environments by mixing up environments or changing room style and object appearances. However, these approaches are still limited by the existing Matterport3D environments. [11] introduces 900 environments from HM3D [46] with generated instructions to address the environment scarcity and learn a more generalizable agent. However, it's expensive to collect 3D environments, and thus their approach is hard to be further scaled up. Different from them, our proposed PANOGEN generates new panoramic environments without the need of any human annotation, and could potentially generate unlimited panoramic environments. Lastly, [24] generates future views based on history observations in a given trajectory so as to build a world model for more informed planning, and [25] synthesizes views at different locations in the original environments as data augmentation for VLN training. Our work focuses on introducing a diverse set of new environments (in appearance), while maintaining consistency in the panorama, using text-conditioned recursive image outpainting.

**Text-to-Image Generation** Text-to-Image generation has been one of the main research areas in both NLP and CV. With the advances of GANs [16], previous works aim to generate high resolution images with stacked generators [60, 61], and improve the perceptual quality of an image by adding regularization losses [6]. However, GANs are hard to optimize [17, 37] and face the problem of mode collapse [38]. Recently, [47] proposes to utilize the latent diffusion models to generate images with high resolution. By first encoding the image into a low dimension latent space that is perceptually equivalent to the pixel space, the latent diffusion model is computationally more efficient while maintaining good generation quality. Diffusion model shows better performance on several image generation tasks like image inpainting [62], image outpainting [59], and image synthesis [42]. [50] first extends the diffusion models to 3D indoor scene synthesis by conditioning it on geometry and editing prompts. Different from them, we propose to generate coherent panorama environments with diverse objects based on detailed text descriptions.

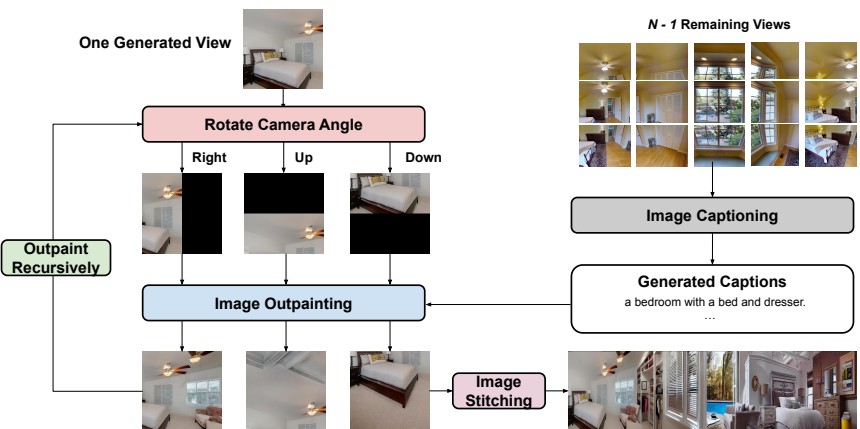

Figure 2: Given an image generated based on the room description, we rotate the camera angle and outpaint the unseen observation recursively to generate a consistent 360-degree panorama view.

## 3 PANOGEN: Generating Panoramic Environments

In this section, we describe our PANOGEN method, which generates panoramic environments from text descriptions for Vision-and-Language Navigation. Our PANOGEN addresses two main challenges for panoramic environment generation. First, the panorama observation of an indoor room environment will contain multiple objects with a complex layout, which makes it hard to use a short description to cover all the details in the panorama. Thus, we first discretize the panorama into 36 views and generate each view separately. However, the separately generated single images will be inconsistent with each other. Hence, to generate consistent panoramic environments, we propose a recursive image outpainting approach (Sec. 3.2). Second, the generated panoramic environment should align with human commonsense knowledge. For example, when generating a panorama for a bedroom, the panorama can consist of objects like "bed", "dresser", but not "refrigerator". To address this problem, instead of utilizing a large amount of available room image captions from the web for different portions in the panorama, which might have unreasonable co-occurrence of objects, we directly generate text descriptions for the panoramas in the Matterport3D environments (Sec. 3.1).

### 3.1 Room Description Collection

We first collect room descriptions from the Matterport3D environments (Figure 2 Right). To maximally generate panorama environments with diverse objects and reasonable layouts, we discretize the panorama into 36 views and caption the discretized views separately. The generated captions contain a more detailed description of the objects in the smaller region of the panorama. We then utilize the pre-trained vision-and-language model from BLIP-2 [32] to caption the discretized room images. BLIP-2 is trained by bootstrapping off-the-shelf pre-trained image encoders and large language models. Benefiting from the general cross-modal alignment between text and image learned during pre-training, and the zero-shot generation ability of large language models, BLIP-2 can generate creative and informative captions for images in the Matterport3D dataset.

### 3.2 Generating Panorama with Text-Conditioned Image Outpainting

While many works [47, 22, 40, 48, 3] have focused on building better models for text-to-image generation, how to transfer the advances in text-to-image generation to text-to-panorama generation is still less explored. [4] combines multiple diffusion generation processes with shared constraints to generate a panorama based on the text description. However, their panorama generation process can only be conditioned on one single text description of the full panorama. One text description is not enough for complex indoor navigation environments with diverse objects. Thus, we propose to generate the panorama view by outpainting recursively based on multiple text descriptions.

As discussed in Sec. 3.1, each discretized image in the Matterport3D environment is paired with a caption generated by BLIP-2. With the detailed descriptions, we use a state-of-the-art text-to-image

diffusion model Stable Diffusion [47] to generate the images. However, the panorama view is not continuous if we directly 'stitch' the generated images. To address this problem (Figure 2), we first generate one single view with zero camera elevation in the panorama based on its caption. The view with zero camera elevation serves as a better starting point, as it usually contains more important information in the panorama, whereas the positive elevation view usually consists of objects on the ceiling, and the negative elevation view usually faces the ground. Next, we rotate the generated image right, up and down by $p_r\%, p_u\%, p_d\%$ respectively, and outpaint the unseen observation based on the caption of the nearby view. By conditioning the image generation on both the text caption and nearby view, the generated images are consistent in style and can be stitched into a coherent panorama. For example, as shown in Figure 2, the caption "a bedroom with a bed and a dresser" mentions "bed", and the bed is already partially generated in the image. When outpainting the unseen observation, our generation model will complete the bed and generate coherent surrounding views, instead of generating a new bed with different appearance. We repeat this process until we generate all 36 views in the panorama. We generate one panorama for each panorama in the R2R training environments. In total, our PANOGEN have 7,644 panoramas, stitched from 275,184 images.

## 4 Utilizing Panoramic Environments for VLN Training

In this section, we first introduce the problem setup of VLN in Sec. 4.1, and the general training procedures for VLN in Sec. 4.2. Then, we introduce the two ways that we utilize the generated panoramic environments. Specifically, we first explore generating new instruction-trajectory pairs in the panoramic environments and utilize these data in VLN pre-training (Sec. 4.3). Then, we directly utilize the generated panoramic environments to replace the original observation during VLN fine-tuning to avoid agents' overfitting to training environments (Sec. 4.4).

### 4.1 Problem Setup

In Vision-and-Language Navigation, the agent needs to navigate through the environment based on natural language instructions. Formally, at each time step $t$, the agent takes in the full language instruction $I$, a panoramic observation $P_t$ of the current location, and navigation history observations $\{P_i\}_{i=1}^{t-1}$. The agent needs to pick the next step from a set of navigable locations $\{g_{t,k}\}_{k=1}^{K}$. We follow the setup in [12], where navigable locations include both adjacent locations which can be reached in one step, and global viewpoints which are observed but not visited in navigation history. Navigation finishes when the agent predicts 'STOP' action or reaches the maximum navigation steps.

### 4.2 VLN Training Procedures

Training a Vision-and-Language Navigation agent contains two stages: pre-training and fine-tuning.

**Pre-training.** In the pre-training stage, the agent is pre-trained with three proxy tasks: Masked Language Modeling (MLM), Instruction and Trajectory Matching (ITM), and Single Action Prediction (SAP). Specifically, in Masked Language Modeling, the agents need to predict the randomly masked words given both the unmasked instructions and the full trajectory observations. In Instruction and Trajectory Matching, the agent needs to learn to pick the correct instruction and trajectory pair from one positive pair and four negative pairs. Two of the negatives come from other trajectories in the same batch, and another two of the negatives shuffle the original trajectories so that the agent could learn the order of the observations in the trajectory. In Next Action Prediction, the agent mimics the navigation task by doing single action prediction based on full instruction and navigation history.

**Fine-tuning.** In the fine-tuning stage, we follow [12] and train the agent with the supervision from the pseudo interactive demonstration. Specifically, at each time step, the ground truth navigation trajectory is sampled based on the current policy learned by the agent. This sampling process enables the agent to explore the environment and generalize better.

### 4.3 Generating Instructions for Paths in Panoramic Environments

Given the high quality and coherent panoramic environments from our PANOGEN, we investigate how to utilize these environments for mitigating the data scarcity problem in VLN agents' training and enhancing VLN agents' generalization ability.

As our PANOGEN environments are generated conditioned on text captions, it shares similar semantics with the original environments, but the room layout and appearance will be different from the original environments. Thus, the instruction for traversing the original environment will not be aligned well with the new panoramic environment. To address this problem, we train a speaker to generate instructions for the new panoramic environments.

Previous works [51, 14, 31, 18] train a small LSTM-based speaker from scratch on VLN datasets to generate instructions for unannotated paths in the Matterport3D environments. However, as these speakers are not pre-trained on larger image-text datasets, they lack general visual grounding knowledge and therefore it's hard to generate instructions with diverse entity mentions that do not appear in the small training data. Marky [54] improves the speaker by utilizing multilingual T5 (mT5) [58], which is a text-to-text encoder-decoder transformer. mT5 enables the speaker to generate instructions with a large multi-lingual vocabulary, and has prior text domain knowledge. However, mT5 only has text domain knowledge, and will need to learn the visual grounding knowledge from scratch on the limited training data. Hence, to introduce general cross-modal alignment information to the agent, we propose to build our speaker based on mPLUG [27], a vision-language model that can do both multi-modal reasoning and generation.

mPLUG is a transformer pretrained on image and text pairs. To adapt it to navigation trajectory, which is a sequence of panorama images, we first simplify the panorama representation to the single view representation which the agent is facing. The single views are first encoded with CLIP-ViT/B-16. The encoded image patches are then flattened and concatenated as the input for instruction generation. We fine-tune mPLUG on the Room-to-Room (R2R) training data, and use it to automatically generate instructions for all the paths in the R2R training data with our new panoramic observation. In total, our speaker data has 4,675 instruction-trajectory pairs.

Following [12], we use R2R dataset [2], and Prevalent dataset [18] for agents' pre-training. Moreover, we further pre-train the agent with our speaker data, which introduces our panoramic environments to the agent and improves the agents' generalization ability.

### 4.4 Learning from Panoramic Environments during Fine-tuning

We further enhance the VLN fine-tuning by incorporating our panoramic environments. Specifically, we randomly replace $m\%$ of the observations in the trajectory with our panoramic environments during fine-tuning. This observation replacement helps the agent avoid overfitting to the limited training environments. As our panoramic environments are generated conditioned on text descriptions of the room, thus the semantic underlying the panoramic observation is similar to the original environments. This ensures that when the replacement ratio $m\%$ is not large, after replacing the original environments with our panoramic observations, the instruction and the observations still have reasonable alignment (discussed in Sec. 6.4).

## 5 Experimental Setup

### 5.1 Dataset and Evaluation Metrics

We evaluate our agent on three datasets: Room-to-Room dataset (R2R) [2], Cooperative Vision-and-Dialog Navigation dataset (CVDN) [52], and Room-for-Room dataset (R4R) [21]. The training set contains 61 different room environments, while the unseen validation set and test set contains 11, and 18 room environments that are unseen during training. Details of the dataset can be found in Appendix.

We evaluate our model on the following metrics: (1) Success Rate (SR), which measures whether the agent stops within 3 meters to the target. (2) Success Rate Weighted by Path Length (SPL) [1], which penalizes long paths that explore the environment to reach the target instead of following the instructions. (3) Goal Progress (GP), which calculates the distance that the agent moves toward the target location. (4) Navigation Error (NE), which is the distance between the stop location and the target. (5) Trajectory Length (TL), which counts the total navigation length of the agent. We consider SPL as the main metric for R2R and R4R dataset, and GP as the main metric for CVDN dataset.

Table 1: Test leaderboard performance for Room-to-Room dataset and CVDN dataset. ♠ indicates approaches that augment the training environments. For "EnvEdit", we report the non-ensemble performance for a fair comparison to all other methods. Best results are in bold, and second best results are underlined.

| Models | Room-to-Room dataset | | | | | | | | CVDN | |
| --- | --- | --- | --- | --- | --- | --- | --- | --- | --- | --- |
| | Validation Unseen | | | | Test | | | | Val | Test |
| | TL | NE↓ | SR↑ | SPL↑ | TL | NE↓ | SR↑ | SPL↑ | GP↑ | GP↑ |
| RecBERT [20] | 12.01 | 3.93 | 63.0 | 57.0 | 12.35 | 4.09 | 63.0 | 57.0 | - | - |
| NDH-Full [23] | - | - | - | - | - | - | - | - | 5.51 | 5.27 |
| HAMT [10] | 11.46 | **2.29** | 66.0 | 61.0 | 12.27 | 3.93 | 65.0 | 60.0 | 5.13 | 5.58 |
| EnvEdit♠ | 12.13 | 3.22 | 67.9 | 62.9 | - | - | - | - | - | - |
| SE3DS♠ [25] | - | 3.29 | 69.0 | 62.0 | - | 3.67 | 66.0 | 60.0 | | |
| DUET [12] | 13.94 | 3.31 | 72.0 | 60.0 | 14.73 | 3.65 | 69.0 | 59.0 | - | - |
| DUET-CLIP | 12.92 | 3.19 | 72.8 | 63.4 | - | - | - | - | 5.50 | - |
| PANOGEN | 13.40 | 3.03 | **74.2** | **64.3** | 14.38 | **3.31** | **71.7** | 61.9 | **5.93** | **7.17** |

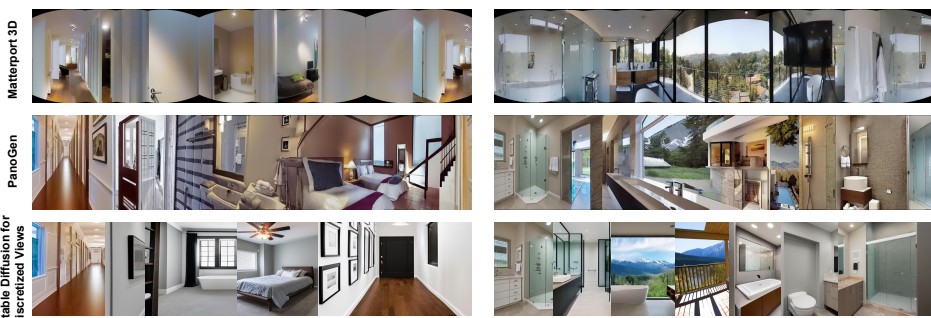

Figure 3: Qualitative analysis of the panoramic environments generated with our PANOGEN. "Matterport3D" is the original environment for VLN tasks. "Stable Diffusion for Discretized Views" is the concatenation of separately generated discretized views given text captions.

## 5.2 Implementation Details

In panoramic environment generation, we caption all the view images in the training environments in R2R dataset with BLIP-2-FlanT5-xxL. We utilize stable-diffusion-v2.1 base model to generate the single view based on caption only, and use stable-diffusion-v1.5-inpainting model to outpaint the unseen observation for the rotated views. In speaker data generation, we build our speaker model based on mPLUG-base. For navigation training, we adopt the agent architecture from DUET [12]. More Implementation Details can be found in Appendix.

## 6 Results and Analysis

In this section, we first present state-of-the-art performance on test leaderboard on Room-to-Room dataset and CVDN dataset in Sec. 6.1. Then, we show some qualitative examples of our generated panoramic in Sec. 6.2. Besides, we further use ablation studies to demonstrate that utilizing our speaker data in the panoramic environments during pre-training, and random replacing the observation with new panorama during fine-tuning are effective for enhancing VLN agents' performance in unseen environments in Sec. 6.3 and Sec. 6.4. Moreover, we investigate how the number of the panorama environments influence the performance in Sec. 6.5, and compare our PanoGen environments with the existing environment augmentation approach in Sec. 6.6. Lastly, we include two quantitative evaluations of the quality of our generated speaker data in Sec. 6.7.

## 6.1 Test Set Results

We show our method's performance on both the Room-to-Room (R2R) and the Cooperative Vision-and-Dialog Navigation (CVDN) dataset. We adopt DUET [12] architecture for our navigation agent.

Table 2: Ablation results for utilizing our speaker data during pre-training on validation unseen set for R2R, CVDN, and R4R datasets.

| Models | Room-to-Room | | | | CVDN | | Room-for-Room | | | |
|---|---|---|---|---|---|---|---|---|---|---|
| | TL | NE↓ | SR↑ | SPL↑ | TL | GP↑ | TL | NE↓ | SR↑ | SPL↑ |
| DUET [12] | 13.94 | 3.31 | 72 | 60 | - | - | - | - | - | - |
| DUET-CLIP | 12.92 | 3.19 | 72.84 | 63.37 | 24.09 | 5.50 | 21.04 | **6.06** | **46.61** | 41.94 |
| PANOGEN+Env-only | 14.21 | 2.99 | 73.35 | 62.12 | - | - | - | - | - | - |
| PANOGEN+EnvDrop | 13.57 | 3.05 | 73.69 | **63.44** | 25.17 | 5.81 | 22.88 | 6.17 | 46.06 | 40.33 |
| PANOGEN+mPLUG | 14.58 | **2.85** | **74.20** | 62.81 | 24.66 | **5.93** | 18.32 | 6.12 | 45.78 | **42.52** |

Different from DUET which uses ViT-B/16 [13] pre-trained on ImageNet to extract features, we use CLIP-ViT-B/16 [45] as it shows better performance on R2R dataset ("DUET" vs. "DUET-CLIP" in Table 1). As shown in Table 1, fine-tuning with panoramic environments from our PANOGEN improves previous SotA agent DUET [12] by 2.7% in success rate, and 2.9% in SPL on Room-to-Room test leaderboard. This demonstrates the effectiveness of utilizing our generated panoramic environments to improve agents' generalization to unseen environments. Moreover, pre-training with our speaker data improves the goal progress by 1.59 meters on CVDN test leaderboard, a relative gain of 28.5% compared with previous SotA agent HAMT [10]. This large improvement demonstrates that learning from our PANOGEN environments are especially helpful for following under-specified instructions in unseen environments which need commonsense knowledge of the visual observation to reach the target. On both the Room-to-Room dataset and CVDN dataset, learning with our PANOGEN environments achieves the new state-of-the-art performance.

## 6.2 Qualitative Analysis of Panoramic Environment

We show some panoramic environment generated with our PANOGEN in Figure 3. We could see that directly generating discretized views based on caption will be disjoint and inconsistent (Row "Stable Diffusion for Discretized Views"). In comparison, our recursive outpainting approach could generate continuous views that can be stitched together to form a high-quality panorama (Row "PANOGEN").

Besides the high quality and coherency, our generated panorama environments is able to preserve the wide range of objects appeared in the original environments, while generating them with new appearance and different room layout. For example, in the left generated panorama, it contains a corridor view, and shows multiple rooms that are connected to the corridor (e.g., bedroom, and bathroom). This layout also follows human's commonsense knowledge, where the bedroom and bathroom can be connected with a corridor. We include more qualitative examples of our panoramic environment and speaker data in Appendix.

## 6.3 Effectiveness of Speaker Data in Pre-training

In this section, we show the effectiveness of utilizing the speaker data generated for our PANOGEN environments for VLN pre-training. As shown in Table 2, pre-training the VLN agent with our speaker data ("PANOGEN+mPLUG") improves the baseline ("DUET-CLIP") by 1.36% in success rate, and 0.2 meters in navigation error on R2R dataset. Besides, we observe 0.43 meters absolute improvement in goal progress (a relative gain of 7.8%) on CVDN validation unseen set. The large improvements in CVDN demonstrates that the agent learns useful commonsense knowledge from the diverse visual environments in our PANOGEN, and thus generalize well to the unseen environments when navigating based on under-specified instructions in CVDN dataset. Pre-training with our speaker data also shows slight improvement in R4R, improving the SPL by 0.58%.

Moreover, we compare our speaker ("PANOGEN+mPLUG") with the widely used speaker from EnvDrop [51] ("PANOGEN+EnvDrop"). We find that utilizing either speaker data improves the baseline performance on R2R and CVDN dataset, demonstrating the effectiveness of our PANOGEN environments for improving agents' generalization to unseen environments. Besides, pre-training with speaker data generated with mPLUG shows better performance in success rate on R2R and R4R dataset and higher goal progress on CVDN dataset, demonstrating the effectiveness of our speaker.

Lastly, we show the effectiveness of generating new instructions for PanoGen environments during pre-training. Compared with only pre-training on original instructions and PanoGen environments

Table 3: Ablation results for replacing the original environment with our panoramic observation during fine-tuning on validation unseen set for R2R, CVDN and R4R datasets.

| Models | Room-to-Room | | | | CVDN | | Room-for-Room | | | |
|---|---|---|---|---|---|---|---|---|---|---|
| | TL | NE ↓ | SR ↑ | SPL ↑ | TL | GP ↑ | TL | NE ↓ | SR ↑ | SPL ↑ |
| DUET [12] | 13.94 | 3.31 | 72 | 60 | - | - | - | - | - | - |
| DUET-CLIP | 12.92 | 3.19 | 72.84 | 63.37 | 24.09 | 5.50 | 21.04 | 6.06 | 46.61 | 41.94 |
| PANOGEN+Replace | 13.76 | **2.99** | **74.41** | **63.88** | **23.29** | **5.63** | 18.62 | **6.02** | **47.78** | **44.25** |

("PANOGEN+Env-only"), we find that generating new instructions with our mPLUG speaker improves the performance by 0.85% in success rate, and 0.69% in SPL on R2R unseen set.

## 6.4 Effectiveness of Panorama Replacement in Fine-tuning

We demonstrate that using the environments from our PANOGEN as observation augmentation during fine-tuning can improve agents' generalization to unseen environments. As shown in Table 3, randomly replacing the observation in the trajectory with our panoramic environments ("PANOGEN+Replace") improves the navigation performance on all the three datasets. Specifically, our approach improves the SR by 1.57% on R2R dataset, 0.13 meters in goal progress on CVDN dataset, and 2.31% in SPL on R4R dataset. The consistent improvements in all the three datasets demonstrate the usefulness of our PANOGEN environments.

We further show that it's important to balance the ratio of replaced observations in the full navigation trajectory. As shown in Table 4, replacing 30% of the viewpoints in the navigation trajectory achieves the best performance. The trajectory will not be aligned with the instruction well if we replace a large ratio of viewpoints with our PANOGEN environments, and thus the improvements in R2R validation unseen set is smaller.

Table 4: Comparison of replacing different ratio of the viewpoints in the trajectory with the panoramic environments generated with our PANOGEN on Room-to-Room validation unseen set.

| No. | Ratio | Validation Unseen | | | |
|---|---|---|---|---|---|
| | | TL | NE ↓ | SR ↑ | SPL ↑ |
| 1 | 0.0 | 12.92 | 3.19 | 72.84 | 63.37 |
| 2 | 0.1 | 13.16 | 3.16 | 72.84 | 63.24 |
| 3 | 0.3 | 13.76 | **2.99** | **74.41** | 63.88 |
| 4 | 0.5 | 13.03 | 3.19 | 72.84 | 63.84 |
| 5 | 0.7 | 12.62 | 3.18 | 72.33 | **63.93** |

## 6.5 Impact of the Number of Panoramic Environments

In this section, we investigate the impact of the number of panoramic environments used during VLN fine-tuning. Specifically, we randomly select 10 scans and 30 scans out of the 61 scans in the R2R training environments. During VLN fine-tuning, if the navigation trajectory belongs to these scans, we replace the original observation with our generated panoramic environments with a fixed probability. As shown in Table 5, we observe that training with more panoramic environments consistently improves the performance (No. 1 - 4). Furthermore, for every panorama in the original R2R training environments, we generate one more panoramic view with our PANOGEN. In this case, the number of panoramic environments we generate is twice the number of the original R2R training environments. As shown in Table 5, we observe that the gain in SPL is still not satured yet (No. 5 vs No. 4), and gradually increases when we add more our PANOGEN environments for VLN fine-tuning. This suggests that it's promising to generate more panoramic environments with our approach to further enhance agents' generalizability.

Table 5: Comparison of replacing the original environments with different number of scans of our panoramic environments.

| No. | # Scans | Validation Unseen | | | |
|---|---|---|---|---|---|
| | | TL | NE ↓ | SR ↑ | SPL ↑ |
| 1 | 0 | 12.92 | 3.19 | 72.84 | 63.37 |
| 2 | 10 | 13.94 | 3.00 | 72.80 | 62.48 |
| 3 | 30 | 13.86 | 3.05 | 73.69 | 62.88 |
| 4 | 61 | 13.76 | **2.99** | **74.41** | 63.88 |
| 5 | 122 | 13.40 | 3.03 | 74.20 | **64.27** |

## 6.6 Comparison with Other Environment Augmentation Approaches

In this section, we demonstrate that training with our PanoGen environments achieves better generalization performance compared to other environment augmentation approaches.

Specifically, we compare our approach with two existing approaches that augment the environments to avoid overfitting: EnvEdit [31] and EnvDrop [51]. For adapting EnvEdit to DUET, we follow the batch mixing approach in EnvEdit and randomly replace half of the data in a batch with the edited environments which change the appearance of the objects during VLN finetuning. For adapting EnvDrop to DUET, we replace the regular dropout layer in DUET with the proposed environment-level dropout layer during fine-tuning. As the results shown in Table 6, training with our PanoGen environments achieves better performance than previous approaches.

Table 6: Comparison of training with different environment augmentation approach on Room-to-Room validation unseen set.

| Model | Validation Unseen | | | |
|---|---|---|---|---|
| | TL | NE ↓ | SR ↑ | SPL ↑ |
| EnvEdit | 13.61 | 3.03 | 72.80 | 63.17 |
| EnvDrop | 13.28 | 3.12 | 72.58 | 62.40 |
| PanoGen | 13.40 | **3.03** | **74.20** | **64.27** |

## 6.7 Quantitative Evaluation of Generated Speaker Data

In this section, we evaluate the quality of our generated instructions by measuring its alignment with the environment, and measuring its BERTScore compared with the original instructions. First, we measure the similarity between the generated instructions and the PanoGen environments. We hypothesize that higher similarity between instruction and trajectory pairs in the embedding space can indicate better alignment between the instruction and the trajectory. Specifically, we represent the trajectory representation by averaging the image embeddings of the viewpoints in the trajectory. The image embedding of each viewpoint is encoded with CLIP-ViT/16. We also encode the instruction with CLIP text encoder. We calculate the cosine similarity between the instruction representation and the trajectory representation. As shown in Table 7, we find that the similarity between our PanoGen environments and instructions generated with mPLUG is higher than instructions generated with EnvDrop (No.2 vs No. 3). Besides, we calculate the similarity between randomly replacing 30% of the viewpoints with PanoGen environments and the original instructions (to mimic the observation replacement fine-tuning). We average the score over 5 runs to mitigate randomness. We find that randomly replacing the observation doesn't lead to decrease in similarity (No. 4 vs No. 1).

Second, we calculate the BERTScore of instructions generated by our mPLUG based speaker and the EnvDrop speaker. Specifically, we use both speakers to generate the instructions on the R2R validation unseen set. We use Bart-base to calculate the BERTScore. Our speaker achieves a BERTScore of 71.8, while the EnvDrop speaker achieves a BERTScore of 70.5.

Table 7: Comparison of similarity between generated instructions and environments.

| No. | Instruction | Environment | Similarity |
|---|---|---|---|
| 1 | Original | Original | 0.2845 |
| 2 | EnvDrop | PanoGen | 0.2669 |
| 3 | mPLUG | PanoGen | 0.2714 |
| 4 | Original | 30% PanoGen, 70% Original | **0.2893** (± 0.0001) |

## 7 Conclusion

In this paper, we propose PANOGEN, a generation approach which can potentially create infinite diverse panoramic environments conditioned on text. Specifically, we propose a recursive image outpainting that reconstructs the missing observations in the panorama gradually based on text caption to generate coherent panorama with diverse objects and room layouts. We then investigate two training methods to effectively utilize the generated panoramic environments during VLN pre-training and fine-tuning. Learning from our PANOGEN achieves the new SotA on both CVDN dataset and R2R dataset, demonstrating its effectiveness for enhancing agents' generalization to unseen environments. **Limitations & Broader Impacts.** See Appendix for limitations and broader impacts discussion.

# 8   Acknowledgement

This work was supported by ARO W911NF2110220, ONR N00014-23-1-2356, DARPA KAIROS FA8750-19-2-1004, and DARPA MCS N66001-19-2-403. The views contained in this article are of the authors and not of the funding agency.

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

# Appendix

In this supplementary, we provide the following:

- Detailed description of the datasets we use in Sec. A, and more implementation details in Sec. B.
- Ablation performance of our PANOGEN on REVERIE dataset in Sec. C.
- Performance of improving panoramic view consistency across different navigation steps in Sec. D.
- More examples of the panoramic environments generated by our PANOGEN in Sec. E, and alignment with speaker data in Sec. F.
- Limitations and broader impacts in Sec. G, and licenses in Sec. H.

## A   Datasets

We evaluate our agent on three datasets: Room-to-Room dataset (R2R) [2], Cooperative Vision-and-Dialog Navigation dataset (CVDN) [52], and Room-for-Room dataset (R4R) [21].

**R2R.** Room-to-Room dataset contains detailed instructions to guide the agents navigate toward the target location step by step. The ground truth paths are the shortest path between the start location and the end location. The training set contains 61 different room environments, while the unseen validation set and test set contain 11, and 18 room environments that are unseen during training.

**R4R.** Room-for-Room dataset is created by concatenating the adjacent paths in the Room-to-Room dataset. In this case, the ground truth path is not the shortest path. This encourages the agent to follow the instructions to reach the target instead of exploring the environment bias and reach the target by directly navigating the shortest path.

**CVDN.** Cooperative Vision-and-Dialog Navigation dataset contains interactive dialogue instructions. The dialogue usually contains under-specified instructions, and the agent needs to navigate based on both the dialogue histories and the commonsense knowledge of the room. The room environments in the training set, unseen validation set, and test set follow the split in Room-to-Room dataset.

## B   Implementation Details

In panoramic environment generation, we caption all the view images in the training environments in R2R dataset with BLIP-2-FlanT5-xxL. We utilize stable-diffusion-v2.1 base model to generate the single view based on caption only, and use stable-diffusion-v1.5-outpainting model to outpaint the unseen observation for the rotated views. It takes 2 days on 6 A100s to generate all the environments.

In speaker data generation, we build our speaker model based on mPLUG-base, which has 350M parameters and utilizes ViT/B-16 as the visual backbone. We train the speaker for 4 epochs on one A6000 GPU with batch size 16 for two days.

For navigation training, we adopt the agent architecture from DUET [12]. We follow the training hyperparameters in DUET. Different from DUET, we utilize CLIP-ViT/B-16 to extract the visual features. We train the model on one A6000 GPU. We pre-train the agent with batch size 64 for 150k iterations, and then fine-tune the agent with batch size 8 for 40k iterations. Both the pre-training and fine-tuning take approximately one day to finish. We report reproduced baseline performance with CLIP-ViT/B-16 features for a fair comparison. The best model is selected based on performance on validation unseen set.

## C   Ablation Performance on REVERIE Dataset

We demonstrate the effectiveness of our approach on REVERIE dataset in Table 8. We observe that pre-training with our speaker data improves the baseline by 2.64% in success rate, while fine-tuning with observation replacement from PanoGen environments achieves 4.60% absolute improvement in success rate. This demonstrates that our approach generalizes well to navigation tasks that have under-specified instructions.

Table 8: Ablation performance on REVERIE unseen set.

| Model | Validation Unseen | | | |
|---|---|---|---|---|
| | SR ↑ | SPL ↑ | RGS ↑ | RGSPL ↑ |
| DUET | 46.98 | 33.73 | 32.15 | **23.03** |
| DUET-CLIP | 46.58 | 34.14 | 31.70 | 22.89 |
| PanoGen+mPLUG | 49.22 | 33.44 | 32.80 | 22.45 |
| PanoGen+Replace | **51.18** | **34.99** | **33.26** | 22.99 |

## D   Improving Consistency Across Steps in Panorama Environment

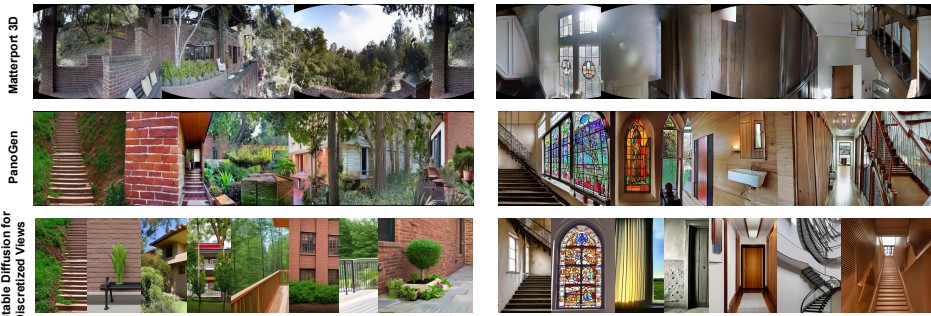

Figure 4: Qualitative analysis of the panoramic environments generated with our PANOGEN. "Matterport3D" is the original environment for VLN tasks. "Stable Diffusion for Discretized Views" is the concatenation of separately generated discretized views given text captions.

In our PANOGEN environments, we didn't explicitly constrain the consistency between views at different navigation steps. In this section, we show our initial exploration of improving the view consistency across steps. Specifically, we fine-tune instructpix2pix [5] based on sub-instruction annotations [19] to generate the next views given previous step in the trajectory. After generating consistent discrete views with instructpix2pix at different viewpoints, we then outpaint the view with our approach to create panoramic environments with better consistency across navigation steps. As shown in Table 9, we observe that

Table 9: Performance of improving consistency across navigation steps in our panoramic environments on Room-to-Room validation unseen set.

| Model | Validation Unseen | | | |
|-------|------|-----|-----|------|
| | TL | NE ↓ | SR ↑ | SPL ↑ |
| DUET-CLIP | 12.92 | 3.19 | 72.8 | 63.4 |
| PanoGen | 13.76 | **2.99** | 74.41 | 63.88 |
| PanoGen+Con | 13.91 | 3.00 | **74.46** | **64.09** |

improving consistency across navigation steps slightly helps with the navigation performance on the R2R unseen set, improving the SPL by 0.21%.

## E    Qualitative Example for PanoGen Environment

We show more panoramic environments generated with our PANOGEN in Figure 4. We observe that directly concatenating discretized views generated separately will generate inconsistent panoramas (Row "Stable Diffusion for Discretized Views"). In comparison, our PANOGEN can generate continuous views with reasonable layout and object co-occurrence (Row "PANOGEN"). Moreover, our approach can generate panorama for both indoor and outdoor environments. Though generating the outdoor environments might not benefit agents' indoor navigation ability directly, our approach demonstrates its potential to be applied to panorama generation with different content (e.g., landscape).

## F    Qualitative Example for Alignment between Speaker Data and PanoGen Environment

We include one panorama-instruction example in Figure 5 to demonstrate the alignment between the instruction and the environment. We show that our panorama environments are more diverse in appearance, and the instruction data generated by our mPLUG based speaker contains more details. Besides, the general semantic information across different steps is still reasonable in consistency.

## G    Limitations and Broader Impacts

Vision-and-Language Navigation tasks can be used in many real-world applications, for example, a home service robot can bring things to the owner based on natural language instructions. In this paper, our proposed method generates panoramic environments for VLN training, and significantly improves navigation agents' generalization ability to unseen environments given limited human-annotated training data. Our approach reduces the efforts of re-training the agents in every new environment when adapting to real-world scenarios.

We also note that there are some limitations of our work. First, this work directly utilizes stable diffusion models trained for inpainting on "laion-aesthetics v2 5+". Though the zero-shot generation performance is good, further improvement might be observed if further trained on room images. Second, we investigate one specific

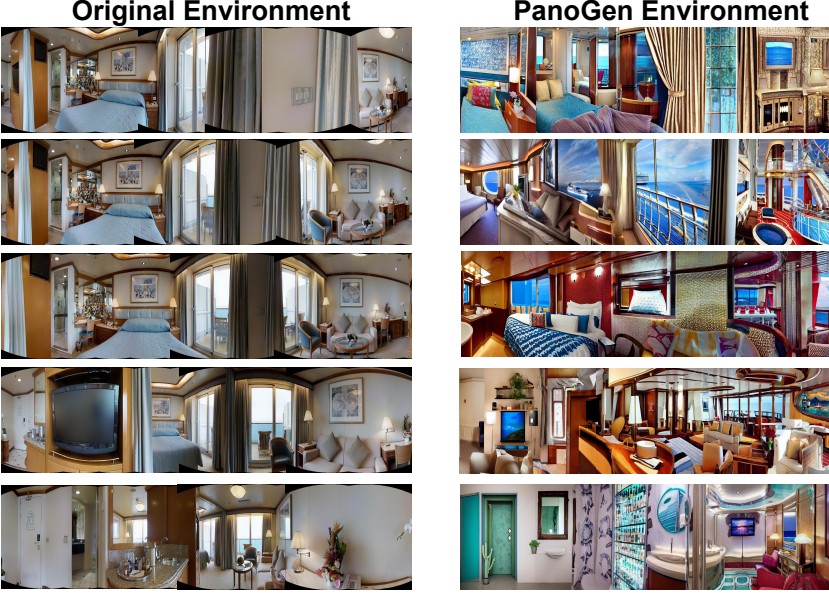

**Original Environment**

**PanoGen Environment**

**Human:** Walk past the TV and continue toward the bathroom. Stop before walking through the bathroom door.

**EnvDrop:** Turn around and walk to the right side of the room. wait there.

**Ours:** walk around the bed and into the bathroom. stop in front of the sink.

Figure 5: Qualitative analysis of panorama-trajectory-instruction pairs.

task Vision-and-Language Navigation in this paper, but the proposed method can be potentially used in other embodied tasks like concept learning and grounding in panoramic environments. We will explore other useful and interesting tasks in the future.

## H   Licenses

We provide the licenses of the existing assets we use in this paper in Table 10.

Table 10: A list of the licenses of the existing assets used in this paper.

| Asset | License |
|---|---|
| Pytorch | BSD-style |
| Huggingface Transformers | Apache License 2.0 |
| Torchvision | BSD 3-Clause "New" or "Revised" License |
| Room-to-Room | MIT |
| Room-for-Room | Apache License 2.0 |
| Cooperative Vision-and-Dialog Navigation | MIT |
| BLIP-2 | BSD 3-Clause "New" or "Revised" License |
| mPLUG | Apache License 2.0 |
| DUET | N/A |
| Stable Diffusion | CreativeML Open RAIL-M |

