# OpenReview forum: "PanoGen: Text-Conditioned Panoramic Environment Generation for Vision-and-Language Navigation"
_NeurIPS.cc/2023/Conference — NeurIPS 2023 poster_

### Official Review · Reviewer_PpHD · 2023-07-02

**Soundness:** 3 good
**Presentation:** 3 good
**Contribution:** 3 good
**Rating:** 5
**Confidence:** 3

**Summary:**

This paper proposes a nove approach for creating an infinite number of diverse panoramic environments conditioned on text for visual-language-navigation (VLN). Specifically, the authors use stable diffusion with captions of images from a existing dataset to generate in-door panoramic views. Recursive inpainting is used to ensure the consistency across views. The authors also present pretraining and finetuning strategies that are designed to maximize the benefits of the synthesized data, and they demonstrate its effectiveness across various benchmarks.

**Strengths:**

1. This paper demonstates that strong generative models (like Stable Diffusion) are potential to augment traning data for robotics learning (VLN in this paper). Similar ideas could be adopted in other areas like autonomous driving.

2. The experiments over different benchmarks show that incorporating virtual panoramic scenes generated by Stable Diffusion is useful.

**Weaknesses:**

1. There are no examples for mPLUG captioning. It would be great to contain several samples of {previous_img, previous_caption, generated_img, generated_caption}. With such kind of samples can reader better understand how mPLUG creates suitable captions for generated images with different layouts and appearances.

2. This method creates novel data with different appearance, scene layouts, and instructions compared to original data. And experiments show that incorporating these three factors together is helpful. However, It is not very clear which specific part or all parts have played a helpful role. If novel layouts and instructions do not help, I believe incorporating ControlNet/GLIGEN can provide novel appearances with the same arrangements, no need for finetuning mPLUG and captioning, making the overall method simpler and easier to follow.

**Questions:**

1. In this paper, panoramic generation is conditioned on captions from existing data. However, it is also possible to create panoramic views in an unconditioned manner. Then mPLUG can also create instructions over this data. Will such data help VLN training?

2. Is it necessary to have novel layouts with instructions, as I mentioned in Weakness (2)? Can we just use ControlNet/GLIGEN?

**Limitations:**

The authors have discussed about the limitations and potential impact in the supp.

---

> ### Author Rebuttal · Authors · 2023-08-09
>
> **Response to Reviewer PpHD**
>
> > Q1: Qualitative examples of generated data.
>
> We include one panorama-trajectory-instruction pair in the general response pdf. As shown in the Figure, our PanoGen environment is more diverse in appearance, but still maintains good alignment in semantics with the original environments. Besides, we find that the instructions generated with our mPLUG based speaker are better in detail compared with the baseline EnvDrop speaker, which clearly indicates that the agent should stop in front of the sink inside the bathroom.
>
> > Q2: Ablations on instructions and layouts.
>
> * To demonstrate the effectiveness of generating new instructions for PanoGen environments, we experiment with pre-training with PanoGen environments and the original instructions. As shown in the Table below, without the instructions generated for the PanoGen environments, we observe 0.8% drop in SR, and 2.2% drop in SPL.
>
> * To demonstrate the effectiveness of layouts, we compare our approach with EnvEdit[1], which only edits the appearance of the original environments by synthesizing based on semantic segmentation (similar idea as using GLIGEN). As shown in the Table below, training with PanoGen environments further improves training with EnvEdit environments by 1.4\% in SR and 1.1\% in SPL.
>
> | Model          | SR   | SPL  |
> |----------------|------|------|
> | PanoGen        | 74.2 | 64.3 |
> | - instructions | 73.4 | 62.1 |
> | - layouts      | 72.8 | 63.2 |
>
> > Q3: Unconditional panorama generation.
>
> Due to time limitations, we cannot train a model to generate a large number of new panoramas unconditionally. Instead, we experiment with randomly switching existing panoramas in PanoGen environments to mimic the unconditional panorama generation process. Then, we use our mPLUG based speaker to generate instructions for these environments. As shown in the Table below, training with this data will decrease the performance by 1.2\% in SR and 1.4\% in SPL. We attribute the performance decrease to the large inconsistency between consecutive viewpoints when using unconditional generation. Conditioning the generation of text captions can implicitly maintain some consistency between consecutive viewpoints.
>
> | Model            | SR   | SPL  |
> |------------------|------|------|
> | PanoGen          | 74.2 | 64.3 |
> | Random switching | 73.0 | 62.9 |
>
>
> [1] EnvEdit: Environment Editing for Vision-and-Language Navigation. In CVPR 2022.

---

> > ### Comment · Reviewer_PpHD · 2023-08-12
> > **Reply to Rebuttal by Authors**
> >
> > I appreciate the clarification provided. I still have a concern. It appears that when training the model using both original instructions and mPLUG-synthesized instructions, the resulting difference in success rate is only 0.8%. While I am not an expert in VLN, based on my previous experience in robotics, I perceive this difference to be relatively insignificant. In order to determine the true contribution of including mPLUG, can you conduct additional experiments using different seeds, and report the mean and standard deviation of the success rate, as this will provide a more comprehensive understanding of how the inclusion of mPLUG impacts the overall performance of the model.

---

> > > ### Author Response · Authors · 2023-08-15
> > >
> > > We're glad you appreciate our clarifications.
> > >
> > > As you suggested, we perform 4 more tests with seeds (1, 42, 1234, 12345). The success rate for each run is shown in the Table below. The mean success rate and std of our PanoGen are 73.59/0.530, while the mean and std of the model trained without synthetic instructions are 72.24/0.804. We do a t-test over the two distributions, which suggests that the two distributions are significantly different with a p-value of 0.008 (i.e., p < 0.01).
> > >
> > > | Seed  | PanoGen     | -instructions |
> > > |-------|-------------|---------------|
> > > | 0     | 74.2        | 73.4          |
> > > | 1     | 73.05       | 72.54         |
> > > | 42    | 73.27       | 71.22         |
> > > | 1234  | 73.31       | 72.03         |
> > > | 12345 | 74.12       | 71.99         |
> > > | Mean  | 73.59       | 72.236        |
> > > | STD   | 0.530424358 | 0.803511045   |
> > >
> > >
> > > We hope these experiments help address your last concern as you mentioned, and hope you could update your score if you are satisfied with the response. Thanks again for your time!

---

> > > > ### Comment · Reviewer_PpHD · 2023-08-18
> > > >
> > > > Thanks for the additional result. Now I understand the difference is significant. However, I still have the concern that <2% improvement in success rate is marginal. Are there any new capabilities / emerging behaviours that help the navigating robot achieve this improvment by incorporating novel instructions? Conducting analyses of this nature would not only provide a deeper understanding of the results but also offer valuable insights beyond the mere reporting of scores. I believe such analyses would enhance the overall impact and interpretation of the study.

---

> > > > > ### Author Response · Authors · 2023-08-20
> > > > >
> > > > > We first want to emphasize that an average 1.4% gain from the generated instructions is statistically significant with a strong p-value < 0.01 (and hence not marginal).
> > > > >
> > > > > Besides, based on your question, we show two new capabilities of our agent: (1) better instruction-following ability (i.e., less backtracking in navigation) and (2) generalization to VLN tasks that have long and dialogue-style instructions.
> > > > >
> > > > > 1. We show that our agent has much less backtracking during the navigation, indicating better instruction-following ability. On the R2R validation unseen set, our agent backtracks 2026 times out of 18506 navigation steps (10.95%), while the agent w/o instruction learning backtracks 2599 times out of 19568 navigation steps (13.28%).
> > > > >
> > > > > 2. The CVDN dataset requires the agent to navigate based on the target object and dialogue history between a navigator and the oracle. On the CVDN dataset, our agent achieves 5.93 Goal Progress (GP), while the agent without learning from new instructions achieves 5.61 Goal Progress (GP).  Goal Progress measures how far the agent moves toward the target location. We achieve an absolute improvement of 0.32 in GP, and a relative improvement of 6.24%. Besides, we want to emphasize that even the ground truth Goal Progress (i.e., the average distance between the start location and end location is 8.36, improving 0.32 in GP indicates a 3.83% improvement towards the optimal navigation.
> > > > >
> > > > > We hope our response addresses your last question, and hope you could update your review accordingly. Thanks again for your time!

---

> > > > > > ### Comment · Reviewer_PpHD · 2023-08-21
> > > > > >
> > > > > > **While a 1.4% gain can be statistically significant, it can still be considered marginal.** To illustrate this point, imagine applying a simple trick to increase the success rate from 90% to 91% without introducing any new insights or capabilities. Such an improvement might be boring.
> > > > > >
> > > > > > However, considering the author's incorporation of new capabilities such as reduced backtracking and improved goal progress, I now feel more satisfied with the 1.4% improvement. Many of my concerns have been addressed, and as a result, I raise my score to 5.

---

> > > > > > > ### Author Response · Authors · 2023-08-21
> > > > > > >
> > > > > > > Thanks for your reply and engagement in the discussion period! We're glad that you appreciate these new capabilities and that many of your concerns are addressed. We'll incorporate your suggestions in the final version.

---

### Official Review · Reviewer_wyXn · 2023-07-05

**Soundness:** 4 excellent
**Presentation:** 4 excellent
**Contribution:** 4 excellent
**Rating:** 7
**Confidence:** 4

**Summary:**

This paper proposed a data augmentation method named PANOGEN, which generates panoramic environments. The proposed method employs a recursive image inpainting technique to generate coherent panoramic environments and incorporates these augmented environments in both the pre-training and fine-tuning stages. Experimental evaluations demonstrate the effectiveness of the proposed method in enhancing performance in VLN tasks.

**Strengths:**

1. PANOGEN generates new environments without human annotation, which is novel and solves the problem of environment scarcity.
2. The paper is well-written and easy to follow.
3. The experiment is sufficient and ablation studies are strict.


**Weaknesses:**

1. It seems that it lacks a discussion about the selection of the image caption model since the caption is a critical section. And I am wondering how it would be to directly generate the panorama view from the instruction.
2. More discussion on the effectiveness of panorama replacement in fine-tuning is needed. I'm still confused about why using the generated environment is better than the original one because of the larger data.
3. It would be better if the visualization results were made into a panorama-instruction format so that readers can compare easily, and there are no failure cases.
4. I could not find any examples of generated instructions or a measure of the quality of the generation, thus it is hard to evaluate the performance of the Speaker.


**Questions:**

It would be better to indicate in Table that DUET-CLIP is the baseline to make it easier to make comparisons.

**Limitations:**

Yes.

---

> ### Author Rebuttal · Authors · 2023-08-09
>
> **Response to Reviewer wyXn**
>
> > Q1: Discussion of selection of image captioning model, and clarification for generating panorama from text.
>
> **Image captioning model choice.** We choose BLIP-2 for image captioning since it has sota/good zero-shot performance on multiple image captioning benchmarks (Table1 in [BLIP-2 paper](https://arxiv.org/pdf/2301.12597.pdf)). Besides, we compare the captions generated by OFA[1] and BLIP-2, and manually go through 50 caption outputs on the R2R seen environments to decide to use BLIP-2.
>
> **Panorama generation from text clarification.** To generate panoramas from text captions, we propose a novel image inpainting approach to gradually generate the panorama from single images (L148-L171). Specifically, each panorama is discretized into 36 views, and we generate captions for each view. Then, we choose one view in the middle elevation as the starting point, and generate the image given the caption. We gradually rotate the image and inpaint the unseen observation recursively to generate the full panorama from multiple text captions.
>
> > Q2: Effectiveness of panorama replacement in fine-tuning.
>
> Due to the limited available training environments for VLN, the previous paper shows that the agent tends to overfit the low-level appearance features of the environment [2]. Thus, training by randomly replacing the observation with PanoGen environments brings more diverse environments during training, and can avoid overfitting and improve generalization (L32-L51).
>
> > Q3: Qualitative example of panorama-instruction pairs.
>
> We include one qualitative example of panorama-instruction pair in the general response pdf. Our PanoGen environments contain similar semantics as the original environments, while being much more diverse.
>
> > Q4: Evaluation of speaker.
>
> **Automatic BERTScore evaluation over instructions.** We calculate the BERTScore of instructions generated by our mPLUG based speaker and the EnvDrop speaker. Specifically, we use both speakers to generate the instructions on the validation unseen set of R2R data. We use Bart-base as the base model to calculate the bert score. Our speaker achieves a BERTScore of 71.8, while EnvDrop speaker achieves a BERTScore of 70.5.
>
> **Qualitative example.** Besides the evaluation with BERTScore, we also include the instructions generated with EnvDrop speaker and our speaker in the general response pdf. We find that our speaker tends to generate instructions with more details, while the EnvDrop instructions only mention “walk to the right side of the room”.
>
> > Q5: Emphasize DUET-CLIP baseline.
>
> Thanks for pointing out. We will add “DUET-CLIP is considered as our baseline approach.” in the Table captions.
>
> [1] OFA: Unifying Architectures, Tasks, and Modalities through a Simple Sequence-toSequence Learning Framework. In ICML 2022.
>
> [2] Diagnosing the Environment Bias in Vision-and-Language Navigation. In IJCAI 2020.

---

> > ### Comment · Reviewer_wyXn · 2023-08-15
> >
> > Thanks for your detailed reply, I have no other concerns.

---

> > > ### Author Response · Authors · 2023-08-18
> > >
> > > Thanks for your reply and positive engagement! We're glad that our response addressed all your questions.

---

### Official Review · Reviewer_Wxjf · 2023-07-05

**Soundness:** 3 good
**Presentation:** 3 good
**Contribution:** 3 good
**Rating:** 8
**Confidence:** 5

**Summary:**

This paper presents a new data augmentation method for VLN tasks. The proposed method first generates captions for each view and then recursively generates new images to ensure the consistency among multi-views. The authors demonstrate two ways to utilize the newly generated panorama on three benchmarks: R2R, CVDN, R4R. Better results are achieved compared to the previous SoTA.

**Strengths:**

1. The paper is well-written and easy to follow, with clearly stated objectives and technical descriptions.
2. Strong performances are achieved on three different VLN tasks.
3. Comprehensive and insightful ablation study is provided.
4. Quantitative and qualitative results show the importance of multi-view consistency to the VLN tasks.

**Weaknesses:**

1. Will the generated panorama in two consecutive steps be largely different? If yes, will it be better if reduce the difference across steps?
2. What are the impacts on high-level VLN tasks, such as REVERIE?
3. It could be better to make comparisons to previous data augmentation methods, such as EnvDrop and PREVALENT.

====== after rebuttal ============
Thanks for the authors' responses. All my concerns have been well addressed.

**Questions:**

See details in weaknesses.

**Limitations:**

See details in weaknesses.

---

> ### Author Rebuttal · Authors · 2023-08-09
>
> **Response to Reviewer Wxjf**
>
> > Q1: Consistency between panorama of two steps.
>
> We show one qualitative analysis in the general response pdf. We could observe that though we didn’t explicitly improve consistency across steps, the general semantic information is still reasonable across steps. For the given example, we could always observe that the navigation happens in a bedroom with a sofa and an outside view of the sea.
>
> > Q2:  REVERIE performance.
>
> We show our approach’s performance on REVERIE in the following Table. Both pre-training with our speaker data and fine-tuning with observation replacement achieve a large performance improvement (4.6% absolute improvement in SR) than the CLIP baseline on the REVERIE dataset.
>
> | Model           | SR    | SPL   | RGS   | RGSPL |
> |-----------------|-------|-------|-------|-------|
> | DUET            | 46.98 | 33.73 | 32.15 | **23.03** |
> | DUET-CLIP       | 46.58 | 34.14 | 31.70 | 22.89 |
> | PanoGen+mPLUG   | 49.22 | 33.44 | 32.80 | 22.45 |
> | PanoGen+Replace | **51.18** | **34.99** | **33.26** | 22.99 |
>
> > Q3: Comparison with EnvDrop and Prevalent.
>
> * **Comparison with PREVALENT**. Our approach is compatible with the existing instruction augmentation approach PREVALENT[1]. As we described in L229-L231, we follow our baseline approach DUET and pre-train the VLN agent on both R2R data and PREVALENT data, which contains synthetic instructions for unannotated paths in the seen environments.
> * **Comparison with EnvDrop**. We adapt EnvDrop to DUET by replacing the regular dropout layer in the image encoder with environment-level dropout. Besides, we also compare with another environment-level data augmentation approach EnvEdit [1], where we follow the batch mixing approach in EnvEdit and randomly replace half of the data in a batch with the edited environments which change the appearance of the objects during VLN finetuning. The results shown in the Table below demonstrate the effectiveness of training with our PanoGen environments.
>
> | Model   | TL    | NE   | SR    | SPL   |
> |---------|-------|------|-------|-------|
> | EnvEdit | 13.61 | 3.03 | 72.80 | 63.17 |
> | EnvDrop | 13.28 | 3.12 | 72.58 | 62.40 |
> | PanoGen | 13.40 | **3.03** | **74.2**  | **64.3**  |
>
> [1] EnvEdit: Environment Editing for Vision-and-Language Navigation. In CVPR 2022.

---

### Official Review · Reviewer_tReg · 2023-07-08

**Soundness:** 2 fair
**Presentation:** 3 good
**Contribution:** 2 fair
**Rating:** 4
**Confidence:** 5

**Summary:**

In this paper, the authors propose to leverage the generative model to create panoramic images for agent training. A recursive inpainting method is adopted to generate 360-degree panorama views, which aims to ensure the co-occurrence of objects and enough diversity. Experiments are conducted on R2R, R4R, and CVDN datasets, confirming the effectiveness of the proposed method.

**Strengths:**

- The proposed pipeline is reasonable.

- The performance improvement is promising when using PANOGEN.

- The paper is well-written and easy to follow.

**Weaknesses:**

- My main concern is the lack of novelty. In my view, the authors simply leverage and modify the existing text2img model to conduct data augmentation for VLN agents, and there is no specific design for the VLN method. Thus I believe this work does not provide enough technical contribution to the VLN community.

- As shown in Table 2 and 3, compared to DUET-CLIP on R2R, the proposed method does not show performance improvement on the SPL.

- More qualitative results of generated panoramic environments need to be provided for better examining the spatiotemporal consistency of panoramas.

- Lacking experiments on datasets with high-level instruction such as REVERIE.

- Missing some references such as [1-4].

[1] HOP+: History-enhanced and Order-aware Pre-training for Vision-and-Language Navigation. In TPAMI.

[2] Adaptive Zone-Aware Hierarchical Planner for Vision-Language Navigation. In CVPR 2023.

[3] LANA: A Language-Capable Navigator for Instruction Following and Generation. In CVPR 2023.

[4] Reinforced Structured State-Evolution for Vision-Language Navigation. In CVPR 2022.

**Questions:**

- Why were the performance results of DUET-CLIP on the test set not provided?

---

> ### Author Rebuttal · Authors · 2023-08-09
>
> **Response to Reviewer tReg**
>
> > Q1: Novelty.
>
> To adapt the text2image model for VLN, our main technical contributions include:
> * We propose a novel inpainting way to generate consistent panorama views for VLN instead of single image views (Sec. 3.2).
> * We propose a multi-modal transformer-based model to generate instructions for our PanoGen environments (Sec. 4.3), which shows superior performance than the baseline speaker when evaluated on downstream tasks, and higher similarity with the PanoGen environments.
> * We propose two ways of effectively incorporating PanoGen environments in VLN training -- (1) utilizing the generated instruction-trajectory pairs in pre-training (Sec. 4.3) (2) replacing observations with augment environments during fine-tuning (Sec. 4.4).
>
> We believe all these aspects are novel and have specific designs suited for VLN tasks, and hence achieve SotA on multiple VLN tasks (i.e., R2R, CVDN, REVERIE). Besides, our approach can be potentially generalized to other text-guided embodied tasks in panoramic environments, and impact a larger research field besides VLN.
>
> > Q2: Performance compared with DUET-CLIP on R2R.
>
> Though having slightly lower SPL on R2R, it achieves significant improvement on CVDN (0.43m absolute improvement in GP, which is a relative 7% improvement in GP). Besides, it also achieves better performance on REVERIE as we showed in Q4.
>
> > Q3: Qualitative results of PanoGen environments.
>
> We provide two more qualitative examples of our PanoGen environments in Appendix Figure 1. Besides, we provide one more panorama trajectory-instruction pair example in the general response pdf for qualitative analysis. Our PanoGen environments contain similar semantics as the original environments, while being much more diverse. The instruction generated by our mPLUG based speaker also contains more detailed instructions.
>
> > Q4: REVERIE performance.
>
> We show our approach’s performance on REVERIE in the following Table. Both pre-training with our speaker data and fine-tuning with observation replacement achieve a large performance improvement (4.6% absolute improvement in SR) than the CLIP baseline on the REVERIE dataset.
>
> | Model           | SR    | SPL   | RGS   | RGSPL |
> |-----------------|-------|-------|-------|-------|
> | DUET            | 46.98 | 33.73 | 32.15 | **23.03** |
> | DUET-CLIP       | 46.58 | 34.14 | 31.70 | 22.89 |
> | PanoGen+mPLUG   | 49.22 | 33.44 | 32.80 | 22.45 |
> | PanoGen+Replace | **51.18** | **34.99** | **33.26** | 22.99 |
>
> > Q5: Missing references.
>
> Thanks so much for pointing out these references. We will definitely add and discuss them in the final version.
>
> > Q6: DUET-CLIP test leaderboard performance.
>
> The test leaderboard only has limited submission times, and the community usually uses it to submit only the final models instead of intermediate baselines (to avoid excessive utilization of the test set and maintain its blind nature). The validation unseen set also contains unseen environments during training to test the generalization ability of our proposed agents, which our agent shows better performance than the baseline approach.

---

### Official Review · Reviewer_VRWy · 2023-07-09

**Soundness:** 4 excellent
**Presentation:** 3 good
**Contribution:** 3 good
**Rating:** 7
**Confidence:** 3

**Summary:**

This paper proposes a creative system-level solution for VL-navigation agent training, by incorporating the recent T2I model to help increase the data diversity while follow the context and human intuition, which facilitate effective pre-training for performing domain-specific tasks. The proposed pipeline with image capturing and recursive in-painting can potentially create an infinite number of diverse panoramic environments conditioned on text. The author has further proposed two ways to demonstrate the usage of this generated panoramic data, by replacing existing images or generating new trajectory-instruction pairs. They have well demonstrated its effectiveness on existing VL-Nav benchmarks, with better generalization to unseen environments.

**Strengths:**

The proposed method of leveraging existing T2I models for VL-Nav tasks is creative and provides a cost-effective way to generate rich data for training embodied agents.

In addition to the effective method of generating panoramas for indoor environments, the author also provides a complete pipeline for using the data and evaluates their idea on well-known public benchmarks. This required considerable effort, but will inspire follow-up work in the community to leverage pre-trained generative models to create diverse training data for downstream tasks.

The proposed framework has achieved exciting results in improving existing methods in instructed navigation tasks, such as improving goal progress by 1.59 meters on the CVDN test set.

The ablation experiments with different ways and ratios of using the generated panorama data are also commendable, as they help to achieve a better system design.

**Weaknesses:**

- It seems that the comparison to previous practices is missing. There have been multiple different practices used to avoid overfitting and increase data diversity, such as manually doing domain randomization, doing re-lighting to existing environments, or doing joint training over multiple different datasets for the same task. If possible, it would be helpful to know how this proposed framework compares to existing practices.

- Another concern is the quality of the synthetic data. While nearby view in-painting might make sense, the final generated panorama can sometimes be counterintuitive and lack loop closure characteristics. The generated trajectory-instruction pair can also sometimes make no sense, and in the replaced view case, the image may differ from the original one a lot regarding the layout. It would be helpful to know if there is a way to measure the quality of the generated data besides the final evaluation in the target tasks, since some generated data might be harmful to the tasks.


**Questions:**

When generating realistic trajectory from the panorama images, will any depth information (like from pre-trained depth) be used as well?

---

> ### Author Rebuttal · Authors · 2023-08-09
>
> **Response to Reviewer VRWy**
>
> > Q1: Comparison with previous practices which avoid overfitting and increase data diversity.
>
> First, our approach is compatible with the existing instruction augmentation approach PREVALENT[1]. As we described in L229-L231, we follow our baseline approach DUET and pre-train the VLN agent on both R2R data and PREVALENT data, which contains synthetic instructions for unannotated paths in the seen environments.
>
> Besides, we further compare our approach with two existing approaches that augment the environments to avoid overfitting: EnvEdit[2] and EnvDrop[3]. For adapting EnvEdit to DUET, we follow the batch mixing approach in EnvEdit and randomly replace half of the data in a batch with the edited environments which change the appearance of the objects during VLN finetuning. For adapting EnvDrop to DUET, we replace the regular dropout layer in DUET with the proposed environment-level dropout layer during fine-tuning.  As the results shown below, training with our PanoGen environments achieves better performance than previous approaches.
>
> | Model   | TL    | NE   | SR    | SPL   |
> |---------|-------|------|-------|-------|
> | EnvEdit | 13.61 | 3.03 | 72.80 | 63.17 |
> | EnvDrop | 13.28 | 3.12 | 72.58 | 62.40 |
> | PanoGen | 13.40 | **3.03** | **74.2**  | **64.3**  |
>
>
> > Q2: Measurement of quality of generated data.
> We measure the quality of our generated data from two aspects.
> * **Similarity between instructions and environments.** First, we measure the similarity between the generated instructions and the PanoGen environments. We hypothesize that higher similarity between instruction and trajectory pairs in embedding space can indicate better alignment between instruction and trajectory. Specifically, we represent the trajectory representation by averaging the image embeddings of the viewpoints in the trajectory. The image embedding of each viewpoint is encoded with CLIP-ViT/16. We also encode the instruction with CLIP text encoder. We calculate the cosine similarity between the instruction representation and the trajectory representation. As shown in the Table below, we find that the similarity between our PanoGen environments and instructions generated with mPLUG is higher than instructions generated with baseline EnvDrop (No.2 vs No. 3). Besides, we calculate the similarity between randomly replacing 30% of the viewpoints with PanoGen environments and the original instructions (to mimic the observation replacement fine-tuning). We average the score over 5 runs to mitigate randomness. We find that randomly replacing the observation doesn’t lead to decrease in similarity (No. 4 vs No. 1).
>
> | No. | Instruction | Environment               | Similarity           |
> |-----|-------------|---------------------------|----------------------|
> | 1   | Original    | Original                  | 0.2845               |
> | 2   | EnvDrop     | PanoGen                   | 0.2669               |
> | 3   | mPLUG       | PanoGen                   | 0.2714               |
> | 4   | Original    | 30% PanoGen, 70% Original | 0.2893 ($\pm$0.0001) |
>
> * **Automatic BERTScore evaluation over instructions.** Second, we calculate the BERTScore of instructions generated by our mPLUG based speaker and the EnvDrop speaker. Specifically, we use both speakers to generate the instructions on the validation unseen set of R2R data. We use Bart-base as the base model to calculate the BERTScore. Our speaker achieves a BERTScore of 71.8, while the EnvDrop speaker achieves a BERTScore of 70.5.
>
> **Qualitative example.** Beside the above automatic evaluation results, we also include one panorama-instruction example in the general response pdf for qualitative analysis. Our PanoGen environments contain similar semantics as the original environments, while being much more diverse. The instruction generated by our mPLUG based speaker also contains more detailed instructions.
>
> **Loop closure characteristics.** Lastly, we are working on improving the loop closure characteristics of our panoramas. Specifically, we generate the last view by conditioning on both its nearby views and inpaint the middle observation. Due to time limitations, we cannot share the results on the VLN task yet, but we will include the results in the final version (and will also try to report it before the rebuttal discussion periods end), as an investigation for how much loop closure characteristics impact the navigation performance.
>
> > Q3: Utilization of depth information.
>
> Depth information is not used in our trajectory generation. We believe utilizing depth information for generating environments that are more consistent in 3D space can be interesting future work.
>
>
> [1] Towards Learning a Generic Agent for Vision-and-Language Navigation via Pre-training. In CVPR 2020.
>
> [2] EnvEdit: Environment Editing for Vision-and-Language Navigation. In CVPR 2022.
>
> [3] Learning to Navigate Unseen Environments: Back Translation with Environmental Dropout. In NAACL 2019.

---

> > ### Comment · Reviewer_VRWy · 2023-08-20
> >
> > Thanks for the explanation and supporting results! My concerns have been well addressed.

---

> > > ### Author Response · Authors · 2023-08-21
> > >
> > > Thanks for your reply and positive engagement! We are glad that our response well-addressed all your questions.

---

### Author Rebuttal · Authors · 2023-08-09

**General Response**

We thank all the reviewers for their thoughtful feedback. We are glad that they find our work novel and creative (Reviewer vRWy, wyXn), and provides a cost-effective way to tackle the data scarcity problem for VLN and potentially more general robotics learning (Reviewer vRWy, wyXn, PpHD). We thank them for recognizing the promising improvement brought by our approach (Reviewer tReg, Wxjf, VrWy), and acknowledging that our ablation studies as comprehensive and insightful (Reviewer Wxjf, wyXn). We thank them for thinking our paper as well-written and easy to follow (Reviewer tReg, Wxjf, wyXn). We address their questions below and will include them in the final version.

---

### Decision · Program_Chairs · 2023-09-21

**Decision:**

Accept (poster)

**Comment:**

The paper proposes a new data generation method for VLN. The initial concerns for the paper mainly include missing comparison with previous data augmentation methods, lacking evaluations on additional datasets, no qualitative examples and marginal performance gain. The rebuttal addressed most of these concerns. All reviewers except reviewer tReg voted to accept the paper. The reviewer tReg did not respond to the authors’ rebuttal, but the AC considered that the clarification on the novelty issue and additional results in the rebuttal are convincing to alleviate the reviewer’s concerns. Therefore, the AC recommends accepting the paper.